# Data Augmentation Techniques to Reverse-Engineer Neural Network Weights from Input-Output Queries

**Alexander Beiser***
TU Wien, Vienna, Austria
alexander.beiser@tuwien.ac.at

**Flavio Martinelli***
EPFL, Lausanne, Switzerland
flavio.martinelli@epfl.ch

**Wulfram Gerstner**
EPFL, Lausanne, Switzerland

**Johanni Brea**
EPFL, Lausanne, Switzerland

**Editors:** Marco Fumero, Clementine Domine, Zorah Lähner, Irene Cannistraci, Bo Zhao, Alex Williams

## Abstract

Network weights can be reverse-engineered given enough informative samples of a network's input-output function. In a teacher-student setup, this translates into collecting a dataset of the teacher mapping – querying the teacher – and fitting a student to imitate such mapping. A sensible choice of queries is the dataset the teacher is trained on. But current methods fail when the teacher parameters are more numerous than the training data, because the student overfits to the queries instead of aligning its parameters to the teacher. In this work, we explore augmentation techniques to best sample the input-output mapping of a teacher network, with the goal of eliciting a rich set of representations from the teacher hidden layers. We discover that standard augmentations such as rotation, flipping, and adding noise, bring little to no improvement to the identification problem. We design new data augmentation techniques tailored to better sample the representational space of the network's hidden layers. With our augmentations we extend the state-of-the-art range of recoverable network sizes. To test their scalability, we show that we can recover networks of up to 100 times more parameters than training data-points.

## 1 Introduction

In the engineering disciplines, "reverse-engineering" aims at revealing the inner processes of systems whose design principles are unknown. In the context of artificial intelligence (AI), successful reverse-engineering of an AI system would pose many risks related to security and privacy of data and models [Oliynyk et al., 2023]. Moreover, understanding how an AI system should be probed would further our fundamental understanding of these systems. Studying how to reverse-engineer neural networks is especially relevant in the context of neuroscience, albeit with some caveats [Marom et al., 2009], where brain circuits are often modelled with neural networks. Here, we set to tackle the non-trivial task of reverse-engineering weights from one-hidden-layer feed forward networks. We make use of a teacher-student setup, where student networks are trained to imitate a teacher network, where the teacher is the model to reverse-engineer. This is done by first probing teacher input-output pairs (querying the teacher), then using this data to train various students that are often wider than the teacher. When a student functionally matches the teacher, the student's solution in weight space is theoretically well understood: student weights either duplicate a teacher neuron, or compute irrelevant features Şimşek et al. [2021]. Generally, achieving functional equivalence

---

*equal contribution.

Proceedings of the III edition of the Workshop on Unifying Representations in Neural Models (UniReps 2025).

by optimizing with gradient descent is not guaranteed to work, since the optimization landscape is high-dimensional and non-convex. "Expand-and-Cluster" (EC) Martinelli et al. [2024] is a recently developed algorithm that leverages theoretical results to identify weights in neural networks. They rely on *over-parameterized* students, i.e. student networks that contain more neurons than the teacher, to allow gradient descent training to reach near-zero loss. After training to functional equivalence, they can extract the original teacher weights from overparameterized students by exploiting the knowledge of the symmetries of the weight space. However, despite the current advances, training students to near-zero loss in practical, large-scale setups still remains challenging. Up to now, teachers having only at most 256 neurons in a single hidden layer can be reconstructed. In this work, we highlight a critical issue that arises when teacher networks are trained on fewer data points than parameters: querying the teacher with its entire training set is not enough to constrain the solution space of the student. Our goal is to understand how many and what type of teacher queries should be performed to ensure students to learn the teacher solution.

**Contributions**. We investigate *empirically* the *quantity* and *quality* of queries needed to train students to functional equivalence. First, we show that with few queries the students tend to overfit. Next, we demonstrate that off-the-shelf data-augmentation methods are also not sufficient to avoid overfitting. Therefore, we designed two data augmentation techniques that allow successful weight recovery from overparameterized teachers. We name them *biased-noise* and *grid composition*. Finally, we report that with our new augmentations we can achieve weight recovery from teachers containing up to 100x more parameters than training data-points[1]

## 2 Related work and experimental setup

Understanding and explaining the inner workings of neural networks is a growing field. Scholars have looked into interpretable activations of neurons in various ways [Olah et al., 2020, Geiger et al., 2021, Wang et al., 2023, Gurnee et al., 2023]. To get the most out of interpretability analysis, researchers often work with simplified tasks. For example, in hierarchical tasks [Petrini et al., 2024], symbolic regression and grokking phenomena [Power et al., 2021, Nanda et al., 2023, Zhong et al., 2024] or under-parameterized networks [Elhage et al., 2022, Simsek et al., 2023]. Another branch focuses on the teacher-student setup: scholars looked at symmetries of equivalent solutions [Petzka et al., 2020, Şimşek et al., 2021, Grigsby et al., 2023] or the structure and geometry of solutions [Tian et al., 2019, Hanin and Rolnick, 2019, Tian, 2020, Martinelli et al., 2025]. This understanding is then leveraged to reverse engineer weights of neural networks [Rolnick and Kording, 2020, Carlini et al., 2020, Fornasier et al., 2022, Canales-Martínez et al., 2024, Martinelli et al., 2024, Foerster et al., 2024, Carlini et al., 2024, 2025, Ito et al., 2025, Chen et al., 2025]. In this work, we focus on improving identifiability methods with data augmentations tailored to sample the most informative signal out of high-dimensional teacher networks.

We explore our augmentations within the Expand-and-Cluster (EC) framework [Martinelli et al., 2024]. We denote weight vectors as $w$ and bias as $b$. For teachers, we write $w^*$ and $b^*$ respectively. The EC pipeline works in four steps, which we discuss in the following:

**Teacher training**. Although not strictly part of EC, we consider the first step of our EC pipeline to be the training of the to-be-imitated teachers. For our experiment setup, we consider teachers trained on MNIST data. The input dimension of this data is $28 \cdot 28 = 784$ with $60,000$ samples to train the teacher. We denote a teacher as $\mathcal{N}$. We consider teachers with one hidden layer and teacher sizes ranging from $4$ to $512$ neurons in the hidden layer. Note that teachers with more than 76 hidden neurons are overparameterised in the sense of having more parameters than datapoints.

**Teacher querying**. The EC algorithm is given the teacher $\mathcal{N}$, which is from the perspective of EC an unknown teacher, i.e., a black box. Given a querying strategy, which is a set of input samples $X$, where $Q = |X|$, EC proceeds to query data from $\mathcal{N}$ resulting in the teacher logits $\mathcal{N}(X)$.

**Student training**. These resulting input-output queries $\mathcal{N}(X)$ are subsequently used to train the students $\mathcal{S}$. We use an overparameterization factor of $\rho = 4$, so students with 4-times as many neurons in the hidden layer as the respective teacher. In detail, students $\mathcal{S}$ are tasked to imitate the teacher logits $\mathcal{N}(X)$ by minimizing a mean square error imitation loss: $\mathcal{L}(X) = \sum_i^Q \frac{1}{Q}(\mathcal{N}(X_i) - \mathcal{S}(X_i))^2$. EC proceeds to train $N$ over-parameterized student networks to fit these queries. Thanks to the overparameterisation, students likely reach near-zero loss.

**Student reconstruction**. Over-parameterized networks learn redundant feature weight vectors that

---

[1]Supplementary material including appendix, code and data is available under: https://github.com/alexl4123/expand-and-cluster.

| Method | $<d(w_i, w_i^*)>$ | $max_i\, d(w_i, w_i^*)$ | $Q$ |
|---|---|---|---|
| MNIST | $7.15 \cdot 10^{-1}$ | $9.03 \cdot 10^{-1}$ | 60k |
| MNIST-Rand.-Rots. | $1.75 \cdot 10^{-2}$ | $8.70 \cdot 10^{-1}$ | 180k |
| MNIST-HVFlips | $1.96 \cdot 10^{-2}$ | $8.74 \cdot 10^{-1}$ | 180k |
| MNIST$\pm\eta_{[-1,1]}$ | $5.04 \cdot 10^{-1}$ | $8.92 \cdot 10^{-1}$ | 180k |
| Grid-Comp. | $5.31 \cdot 10^{-3}$ | $5.77 \cdot 10^{-1}$ | 180k |
| Grid-Comp.$\pm\eta_{[-1,1]}$ | $3.24 \cdot 10^{-1}$ | $8.76 \cdot 10^{-1}$ | 180k |
| MNIST$\pm\eta_{[0,1]}$ | $\mathbf{2.53 \cdot 10^{-8}}$ | $\mathbf{3.34 \cdot 10^{-7}}$ | 180k |
| Grid-Comp.$\pm\eta_{[0,1]}$ | $\mathbf{9.74 \cdot 10^{-6}}$ | $\mathbf{2.01 \cdot 10^{-4}}$ | 180k |

Table 1: Results of augmentation techniques on reconstruction. $<d(w_i, w_i^*)>$ and $\max_i d(w_i, w_i^*)$ denote the average and maximum cosine distance between teacher and reconstructed student neurons, respectively. $Q$ is the dataset size.

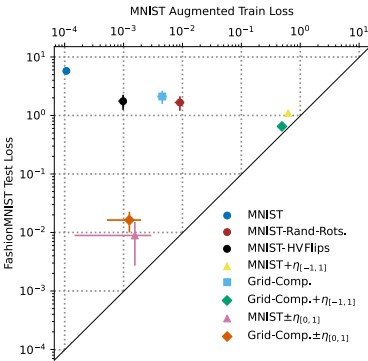

Figure 1: Scatter plot between train and FashionMNIST-test datasets of augmented datasets. Loss measured as MSE.

are filtered out by a clustering procedure. The reconstructed network is obtained by collapsing each cluster into a single neuron, followed by a final fine-tuning phase on the teacher queries. It was shown that *if these students are functionally equivalent to the teacher (for any input)* the student neurons copy the teacher neurons weights modulo various symmetries [Şimşek et al., 2021]. These symmetries are due to neuron permutations, properties of the activation function and over-parameterization. For simplicity and best numerical accuracy, throughout this paper we consider the asymmetric activation function $\mathrm{g}(x) = \mathrm{softplus}(x) + \mathrm{sigmoid}(4x)$. The same principle applies to all standard activation functions explored in Martinelli et al. [2024]

## 3 Results

We proceed with our results and the discussion thereof. Table 1 and Figure 1 show the results of reconstructing a teacher with 512 neurons in the hidden layer, with data augmentation teacher querying strategies. Figure 2 shows a figure of measured pre-activation variability for different augmentation techniques and our scalability study w.r.t. teacher size when restricting the available base dataset for the augmented teacher queries.

### 3.1 The overfitting problem in reconstruction

The original Expand-and-Cluster (EC) work shows that parameter recovery is possible for teachers with up to 256 neurons in the hidden layer, when trained on the MNIST dataset ($Q = 60k$) [Martinelli et al., 2024]. To push the EC boundaries and recover networks of larger sizes, we experimented with increasing the hidden layer size from 256 to 512 and followed the EC procedure described above. We report comparable training losses (MSE) of $\approx 10^{-6}$ for the 256-, and $\approx 10^{-4}$ (Figure 1, blue "MNIST" circle) 512-teachers. However, we fail in the reconstruction of the 512 teacher (Table 1, first row). We conclude that the training loss *is not* a good predictor of reconstruction success. We hypothesize that an out-of-distribution test loss will be a good indicator whether the student successfully imitates the teacher and does not overfit on the training data. We chose the (training) FashionMNIST dataset as an out-of-distribution test dataset (Figure 1). In the appendix, we additionally show the behavior of the (test) FashionMNIST (Figure 5) and the (test) MNIST (Figure 5) datasets as test datasets. On the (training) FashionMNIST dataset, we observe orders of magnitude higher (test) losses compared to the MNIST training dataset ($\approx 5 \cdot 10^0$ vs. $\approx 10^{-4}$, MNIST Figure 1). We speculate that the number of queries $Q = 60k$ is too low to constrain the students to align to all the 407k parameters of the teacher; this leads the students to *overfit* on the training data. We confirm our hypothesis by observing failed parameter recovery of the teacher (MNIST Table 1, first row).

### 3.2 The failure of standard augmentations

Given the need for more input samples to query the teacher, we first explore some standard augmentation techniques used in deep learning. In the setting of Figure 1, we aim to recover the weights of a one hidden-layer teacher network of 512 hidden neurons trained on MNIST. We augment the MNIST dataset with randomly rotated images (MNIST-Rots.), horizontal and vertical flipping

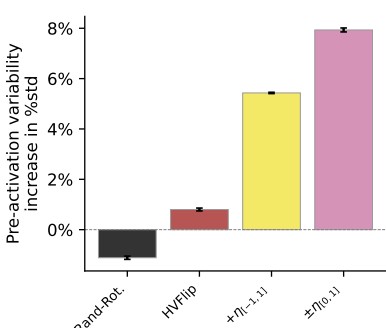
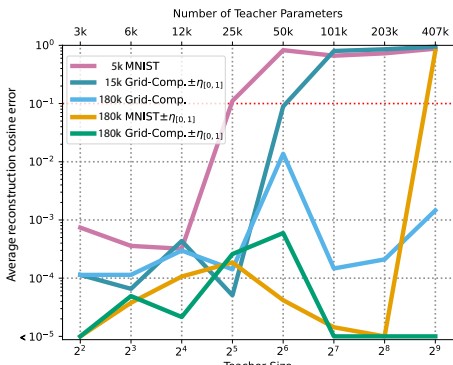

Figure 2: Pre-activation variability increases for different augmentation techniques, compared to the variability of MNIST. Error bars indicate the standard error of the mean (left). Average cosine distances for teacher sizes $2^2$ to $2^9$ trained on $5k$ MNIST data-points (right). Every student is only trained on queries from the same $5k$ MNIST subset plus augmentations.

(MNIST-HVFlips) and by adding uniform independent noise ($\eta_{[-1,1]} \sim \mathcal{U}[-1,1]$) to each pixel of every image in the training set (MNIST $\pm\eta_{[-1,1]}$). The new augmentations increase the dataset size from $60k$ to $180k$. Although relatively good training losses are achieved for rotation and flipping, evaluating the trained students on FashionMNIST results in poor generalization losses (Figure 1). Adding uniform random noise led to failure in both training and evaluation loss. This suggests that the students did not learn to imitate the teacher. We show the failure in reconstruction in Table 1.

### 3.3 Inducing rich activations in teacher neurons

We hypothesize that the failure of standard augmentation techniques stems from *little to no variability* in teacher pre-activation. Our input dimension is $784$, therefore it is likely that classically augmented data $x_{\text{aug}}$ provide almost no variation in pre-activation: $w^* \cdot x_{\text{aug}} \approx 0$. This phenomenon, which does not arise in low dimensional settings, relates to non-informative teaching signals for the students. We think that variability along the teacher hidden weight vectors is a key feature for constructing a good augmented dataset. Ideally, in a one-dimensional setup, three data-points spanned along the feature weight vector suffice to fit the nonlinear activation of a single hidden teacher neuron (Figure 3A). But, of course, we do not know the weight vector since it is the object we are trying to infer. The magnitude of the variability should be chosen carefully: too high magnitude might degenerate to cases where the post-activation of the perturbed data-point lies in the asymptotic regimes of the activation function. This results in data-points lying far away from the nonlinear part of the activation function (Figure 3B).

By moving to higher-dimensional inputs, we find inspiration from Tian [2020]. Their (practically unfeasible) sampling strategy for ReLU teacher identification consists of perturbing input data-points along all input dimensions with a magnitude large enough to have augmented data on both sides of the teacher hyperplane (i.e. the subspace spanned by $w^* \cdot x + b = 0$). This aligns with our intuition about having variability in the pre-activations, as shifting points between sides of hyperplanes amounts to a perturbation along the teacher weight vector. In our setup, we consider additive augmentations to our inputs, $x_{\text{aug}} = x + a$, and write the pre-activation of the neuron as $w^* \cdot (x + a) + b^*$.

We now explore how different augmentation techniques affect the variability of pre-activations: given our previous arguments, the most informative samples should induce variation in the teacher pre-activation and therefore, should (intuitively) be close to the teacher training data. Nevertheless, our observations show that adding zero-mean uniform noise, $\eta_{[-1,1]}$, to such samples does not induce a high variability of pre-activations (Figure 2), resulting in failure of reconstruction (Table 1). To promote variability of pre-activations, we *introduce biased noise augmentations*: for each image in the training set we create two sets of perturbed images, where the noise for each pixel of an image in the first set is sampled independently from $\eta_{[0,1]} \sim \mathcal{U}[0,1]$ and for images in the second set from $\eta_{[-1,0]} \sim \mathcal{U}[-1,0]$. We abbreviate this augmentation procedure with $\pm\eta_{[0,1]}$. We hypothesize that a mix between perturbation along a fixed direction combined with random perturbations is a good augmentation strategy, which we confirm by our experiments, as biased noise augmentation $\pm\eta_{[0,1]}$ leads to excellent results (Figure 1 and Table 1). Regarding the magnitude of our perturbation, we

empirically choose a value of 1, as the standard deviation of scaled MNIST is 1. Our results indicate that, indeed, a value of 1 yields good performance in practice (Figure 1 and Table 1).

### 3.4 How far can we augment the original data?

We proceed to show that our techniques are able to reconstruct networks of up to $100\times$ more parameters than input samples. We aim at reconstructing teacher networks of different hidden layer sizes (from $2^2$ to $2^9$) trained on a fraction of MNIST with just $5k$ samples. For different augmentation techniques, we show our results in Figure 2. Without data augmentation, it is only possible to recover teachers of up to 32 hidden neurons. With the biased-noise augmentation, MNIST $\pm \eta_{[0,1]}$ to $180k$ samples we observe that we are able to recover teachers of hidden layer sizes of up to *256*-neurons, but reconstruction fails for the *512*-teacher (Figure 2). We attribute the failure for the *512*-teacher to the fact that the augmentation of the dataset from $5k$ to $180k$ with simple noise addition might lead to non-informative images. For this reason, we introduce *Grid-Composition* (Grid-Comp.), a method that combines MNIST images to form new data, which is similar to MNIST. In detail, we construct new images by stitching together 9 image patches into a 3x3 grid layout. With image-composition we can sample up to $D^9$ data-points that are similar to MNIST, where $D$ is the amount of available images. Although we achieve good results also with simple Grid-Comp., thereby recovering most neuron weight vectors, we fail to reconstruct all neuron weight vectors (see Appendix for details). We denote the combination of biased noise and Grid-Composition as Grid-Comp. $\pm \eta_{[0,1]}$. For our experiments, we chose $60k$ data-points of Grid-Comp. and then add equally sized positive, and negative biased noise parts. With Grid-Comp. $\pm \eta_{[0,1]}$ we are able to reconstruct both the $2^9$ sized teacher trained on 5k MNIST images, as shown in Figure 2, as well as the teacher trained on the full 60k MNIST dataset (Figure 1).

## 4  Conclusion

Previous methods [Martinelli et al., 2024] were unable to reconstruct teachers that have a relatively large amount of parameters when compared to the number of training data-points available. To overcome this problem, we develop data-augmentation techniques tailored to elicit enough variability in teacher pre-activations. Our experiments confirm our hypothesis by being able to identify weights in previously unreconstructible teachers in the MNIST setting. We report excellent alignment between student and teacher weights for our biased-noise augmentation, while standard data augmentation methods used in the field fail. Moreover, we showed that we can push the limits of our augmentation technique and reconstruct teachers of up to 100x more parameters than training data-points, where we use an augmentation technique which has a dataset size of 35 times the original size. We believe our methodology will be useful to scale the field of reverse engineering network weights to more complex network architectures and higher input dimensions.

# 5 Acknowledgements

Alexander Beiser conducted the research during the summer@EPFL 2024 program. This research was supported by Frequentis and by the Swiss National Science Foundation grant CRSII5 198612.

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

# A  Appendix

**Limitations:**  We stopped our scaling experiments of reconstruction at teacher sizes of 512 neurons in the hidden layer, due to computational requirements. Reconstruction of larger teachers, like 1024 neurons, fails due to a lack of available RAM on our benchmark servers. We work towards improving the algorithm s.t. the reconstruction requires less RAM. Our results are, at the moment, strictly empirical. We are working towards a rigorous theoretical framework of teacher reconstruction, which includes our observed problems of overfitting.

**Experimental System:**  We used Expand-and-Cluster as the practical tool for reconstructing teachers. Library wise, we used the *PyTorch 24.01*[2] container release. Hardware wise, we used *NVIDIA Tesla V100 32G* and *NVIDIA Tesla A100 40G* GPUs, combined with *Intel Xeon Gold 6240* and *AMD EPYC 7302* CPUs respectively, and up to 512GB of RAM.

For Expand-and-Cluster we use parameters of $N = 30$, $\gamma$ ranging from $0.5$ to $0.75$, and $\beta = 3$. We used adam optimizer for training and tuning. Further, we used a plateau learning rate scheduler. We measure training time in iteration steps ($\text{Steps} = (\#\text{EPOCHS} \cdot |X|)/|\text{Batch}|$), as we have datasets with differing size and therefore epochs are misleading (see Fig. 6 and Fig. 7).

| Method | r | N | m/r | $<d(w_i,w_i^*)>$ | $max_i d(w_i,w_i^*)$ | $<d(a_i,a_i^*)>$ | $max_i d(a_i,a_i^*)$ | Q |
|---|---|---|---|---|---|---|---|---|
| MNIST | **512** | 30 | 0.16 | $7.15 \cdot 10^{-1}$ | $9.03 \cdot 10^{-1}$ | $4.03 \cdot 10^{-1}$ | $1.36 \cdot 10^{0}$ | 60k |
| MNIST-Rand-Rots. | **512** | 30 | 1.04 | $1.75 \cdot 10^{-2}$ | $8.70 \cdot 10^{-1}$ | $1.20 \cdot 10^{-2}$ | $1.28 \cdot 10^{0}$ | 180k |
| MNIST-HVFlips | **512** | 30 | 1.02 | $1.96 \cdot 10^{-2}$ | $8.74 \cdot 10^{-1}$ | $9.85 \cdot 10^{-3}$ | $8.95 \cdot 10^{-1}$ | 180k |
| MNIST$\pm\eta_{[-1,1]}$ | **512** | 30 | 0.41 | $5.04 \cdot 10^{-1}$ | $8.92 \cdot 10^{-1}$ | $2.55 \cdot 10^{-1}$ | $1.29 \cdot 10^{0}$ | 180k |
| Grid-Comp. | **512** | 30 | 1.08 | $5.31 \cdot 10^{-3}$ | $5.77 \cdot 10^{-1}$ | $3.24 \cdot 10^{-3}$ | $5.52 \cdot 10^{-1}$ | 180k |
| Grid-Comp.$\pm\eta_{[-1,1]}$ | **512** | 30 | 0.64 | $3.24 \cdot 10^{-1}$ | $8.76 \cdot 10^{-1}$ | $1.63 \cdot 10^{-1}$ | $1.39 \cdot 10^{0}$ | 180k |
| MNIST$\pm\eta_{[0,1]}$ | **512** | 30 | 1.00 | $2.53 \cdot 10^{-8}$ | $3.34 \cdot 10^{-7}$ | $1.26 \cdot 10^{-8}$ | $1.16 \cdot 10^{-7}$ | 180k |
| MNIST$\pm\eta_{[0,2]}$ | **512** | 30 | 1.00 | $1.51 \cdot 10^{-2}$ | $8.7 \cdot 10^{-1}$ | $8.32 \cdot 10^{-1}$ | $9.24 \cdot 10^{-1}$ | 180k |
| MNIST$\pm\eta_{[0,0.5]}$ | **512** | 30 | 1.00 | $3.58 \cdot 10^{-1}$ | $8.77 \cdot 10^{-1}$ | $1.84 \cdot 10^{-1}$ | $1.45 \cdot 10^{0}$ | 180k |
| Grid-Comp.$\pm\eta_{[0,1]}$ | **512** | 30 | 1.00 | $9.74 \cdot 10^{-6}$ | $2.01 \cdot 10^{-4}$ | $4.94 \cdot 10^{-7}$ | $9.88 \cdot 10^{-6}$ | 180k |

Table 2: Reconstruction of 512-hidden-neuron teacher with $g$-activation function for different data augmentation methods. $r$ is the hidden layer-teacher size. $N$ is the number of students. $m/r$ is the fraction of neurons reconstructed vs. hidden layer size of the teacher. $<d(w_i,w_i^*)>$ and $max_i d(w_i,w_i^*)$ denote the average and maximum cosine distance between teacher and reconstructed student weight vectors, respectively. $<d(a_i,a_i^*)>$ and $max_i d(a_i,a_i^*)$ denote the average and maximum cosine distance between teacher and reconstructed student output weights, respectively. MNIST-HVFlips, MNIST-Rand-Rots. fare better, but are still unable. Trivial overlay (MNIST$\pm\eta_{[-1,1]}$) and (Grid-Comp.$\pm\eta_{[-1,1]}$) also fails in reconstruction. MNIST fails to recover the teacher. Grid-Comp. fares slightly better, but still falls short in reconstruction. Further, MNIST$\pm\eta_{[0,2]}$ and MNIST$\pm\eta_{[0,0.5]}$ fail to reconstruct the teacher. Grid-Comp.$\pm\eta_{[0,1]}$, and MNIST$\pm\eta_{[0,1]}$ are able to reconstruct the teacher.

**Additional Experimental Data and Results:**  In Figure 3 we show our intuition about a correct fit of variability and our *limitations*. A correct fit should not have too high magnitudes of variability (green datapoints), but should constrain the activation function (red datapoints).

Regarding limitations, we are aware that *only* looking at teacher pre-activations does not suffice in high-dimensionality cases in general. We show an example, where pre-activations do not suffice for the ReLU activation function in a low-dimensional (2D) setting in Figure 3. There, data was sampled in an unfortunate way, leading to a case where just 1 student neuron (green) suffices to functionally simulate 2 teacher neurons (red). The arrows indicate the weight vector $w$, where the thick line indicates the hyperplane. The thin red lines show the combined output of the 2 teacher neurons. The blue dots represent the sampled data.

In Figure 4 we show an example how a datapoint is constructed for our Grid-Comp. and Grid-Comp.$\pm\eta_{[0,1]}$ augmentation techniques.

In Figure 5 we show additional loss plots between the training and different test losses (a,b). In (a) we compare the loss of the MNIST-train dataset to the Fashion-MNIST-test dataset and in (b) we compare the loss of the MNIST-train dataset to the MNIST-test dataset. In (c) we show the maximum

---

[2]Pytorch 24.01

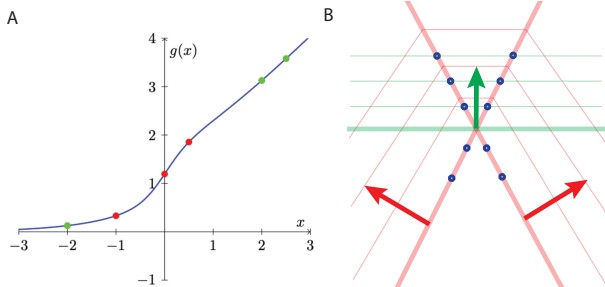

Figure 3: (A): Data-points projected along teacher weight vector: low variability ensures good fit (red points) while too high variability samples the asymptotic part of the activation function leading to a bad fit (green). (B): ReLU example where variability alone is not enough. Data variability for both teacher vectors (red), but a single student vector fits the data perfectly (green).

cosine distance (compared to the average cosine distance shown in the main part) of reconstruction, when increasing teacher sizes.

In Figure 6 we show the training behavior when the student is trained on the teacher queries for a subset of $Q = 5k$ MNIST datapoints. We compare the train loss ($Q = 5k$) to two test losses: On the full MNIST dataset ($Q = 60k$) and on the FashionMNIST train dataset ($Q = 60k$). In (a,b,c) the training behavior is shown for a teacher comprising 8 neurons. (a) shows the behavior on the train loss, (b) on the full MNIST dataset, and (c) on the FashionMNIST dataset. (d,e,f) shows the behavior on a teacher comprising 32 neurons. (d) shows the behavior on the train loss, (e) shows the behavior on the full MNIST dataset, and (f) shows the behavior on the FashionMNIST dataset. Figure 7 extends the results shown in Figure 6 to the 128 and 512 teacher cases.

Finally, Figure 8 shows distributions of pre-activations of MNIST compared to different data augmentation techniques.

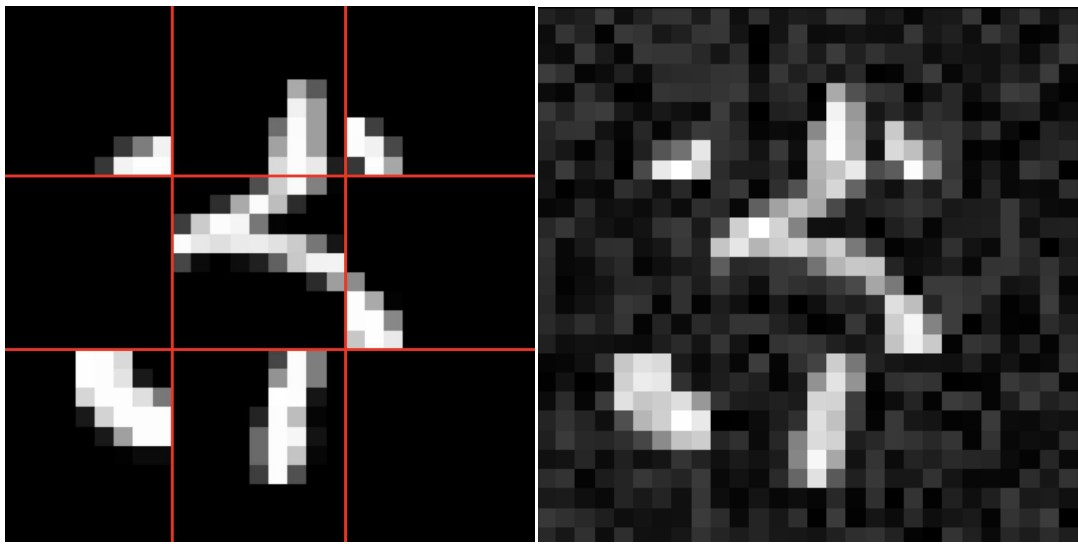

(a) Grid-Comp. sampling, with $c = x \cdot y$, where $x = 3$ and $y = 3$.

(b) Grid-Comp.$\pm \eta_{[0,1]}$ example.

Figure 4: Visualization of the Grid-Comp. data augmentation technique.

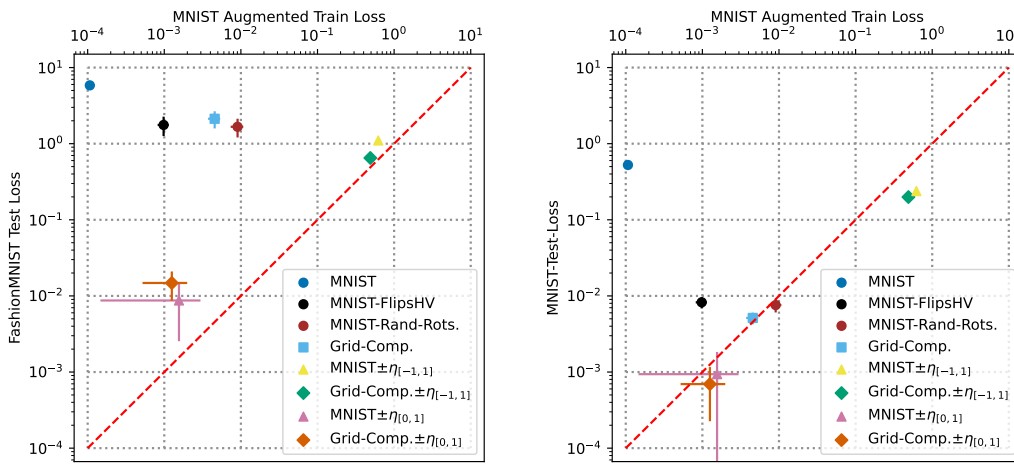

(a) Comparison of losses between training and Fashion-MNIST Test dataset.

(b) Comparison of losses between training and MNIST Test dataset.

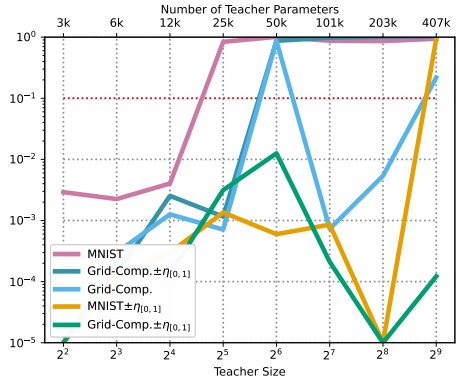

(c) Maximum cosine distances for teachers on 5kMNIST, with teacher sizes ranging from $2^2$ to $2^9$.

Figure 5: (log)-loss plots for augmentation techniques of the teacher with 512 hidden neurons, which was trained on the MNIST dataset (a), (b) and (c). (d): Reconstruction of data augmentation techniques on the 5k MNIST fragment.

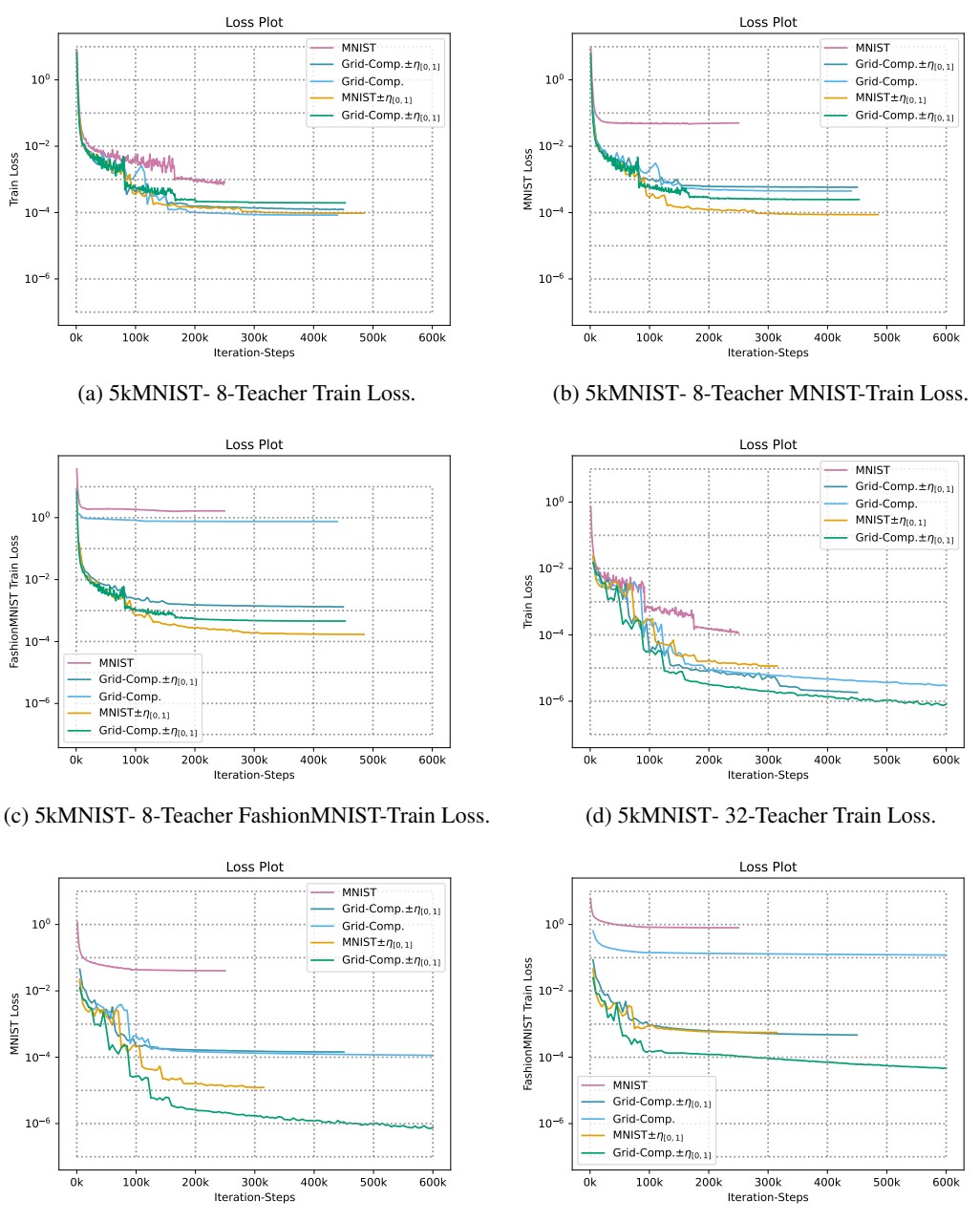

(a) 5kMNIST- 8-Teacher Train Loss.

(b) 5kMNIST- 8-Teacher MNIST-Train Loss.

(c) 5kMNIST- 8-Teacher FashionMNIST-Train Loss.

(d) 5kMNIST- 32-Teacher Train Loss.

(e) 5kMNIST- 32-Teacher MNIST-Train Loss.

(f) 5kMNIST- 32-Teacher FashionMNIST-Train Loss.

Figure 6: Loss plots of various teacher sizes. Their comparison show overfitting behavior when contrasting train loss to MNIST-Train loss, or FashionMNIST-train loss. Contrast for this (a) to (b) and (c), and (d) to (e) and (f).

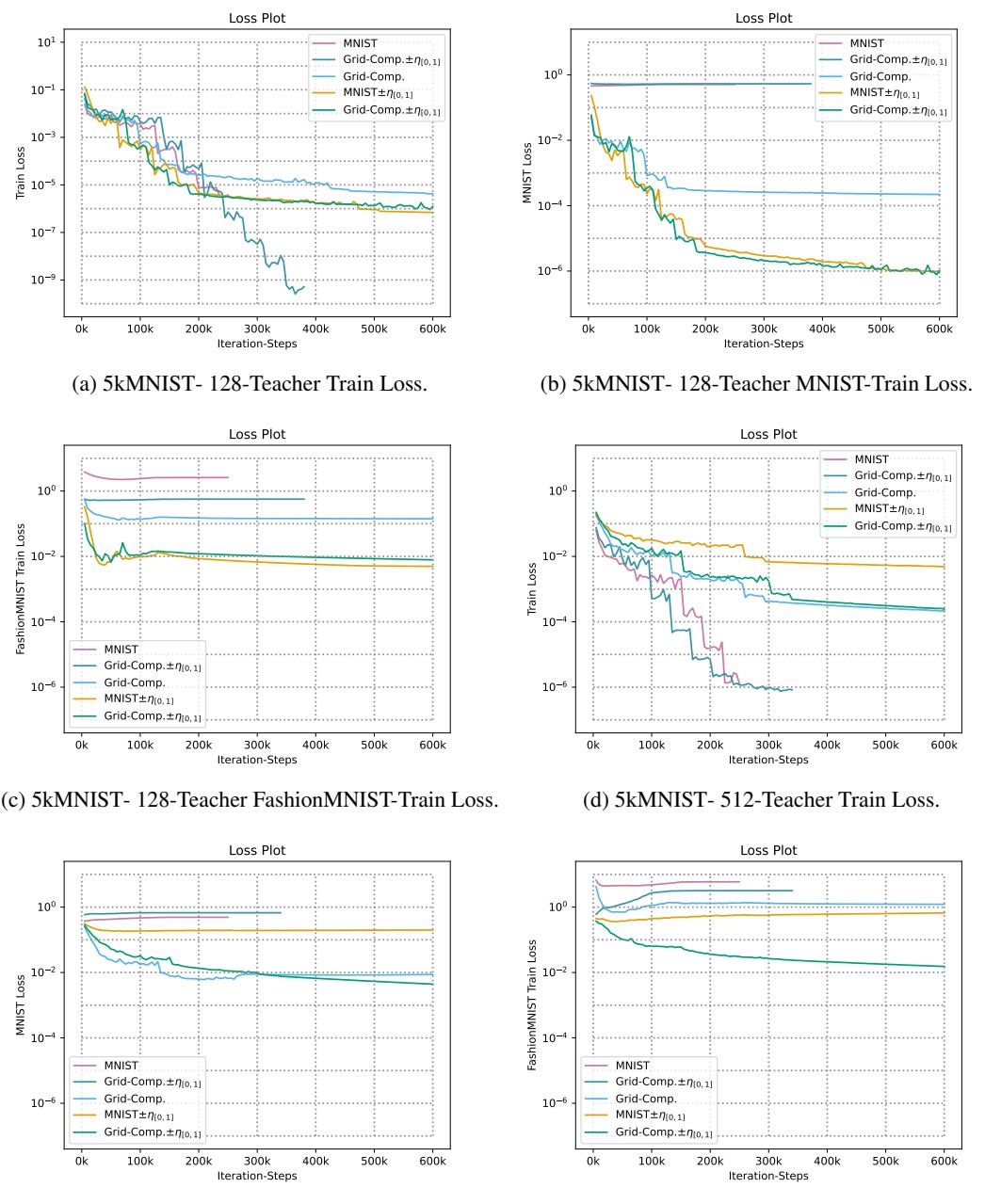

(a) 5kMNIST- 128-Teacher Train Loss.

(b) 5kMNIST- 128-Teacher MNIST-Train Loss.

(c) 5kMNIST- 128-Teacher FashionMNIST-Train Loss.

(d) 5kMNIST- 512-Teacher Train Loss.

(e) 5kMNIST- 512-Teacher MNIST-Train Loss.

(f) 5kMNIST- 512-Teacher FashionMNIST-Train Loss.

Figure 7: Loss plots of various teacher sizes. Their comparison show overfitting behavior when contrasting train loss to MNIST-Train loss, or FashionMNIST-train loss. Contrast for this (a) to (b) and (c), and (d) to (e) and (f).

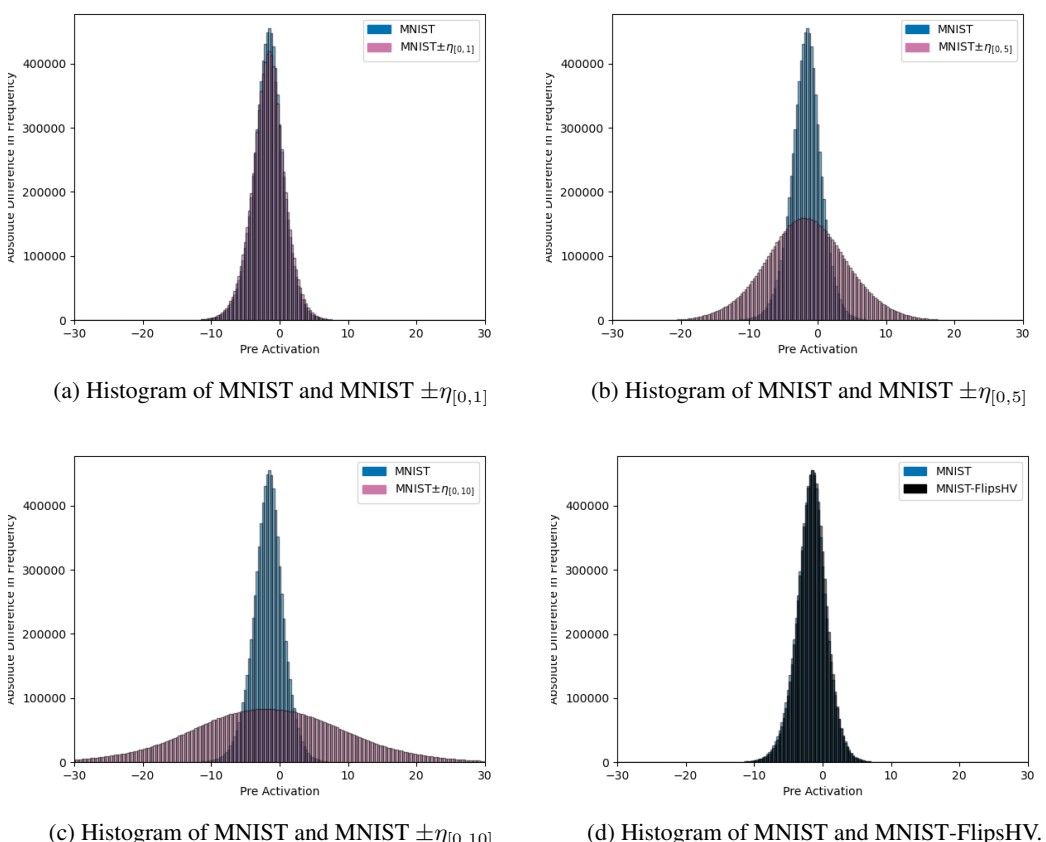

(a) Histogram of MNIST and MNIST $\pm\eta_{[0,1]}$

(b) Histogram of MNIST and MNIST $\pm\eta_{[0,5]}$

(c) Histogram of MNIST and MNIST $\pm\eta_{[0,10]}$

(d) Histogram of MNIST and MNIST-FlipsHV.

Figure 8: Histograms of neuron pre-activations ($w \cdot x + b$) neurons of a teacher with 512 hidden neurons, comparing MNIST to MNIST $\pm\eta_{[0,u]}$ for $u \in \{1, 5, 10\}$, and to MNIST-FlipsHV. An increase in $u$ leads to a widening in the variability. We see a slight sharpening in variability for MNIST-FlipsHV.

