# OpenReview forum: "Data Augmentation Techniques to Reverse-Engineer Neural Network Weights from Input-Output Queries"
_NeurIPS.cc/2025/Workshop/UniReps — UniReps2025_

### Official Review · Reviewer_JxR5 · 2025-09-14
**Clever augmentations (biased-noise + grid composition) boost EC weight recovery; clarify neuron matching/metrics and add fixed-Q ablations (Weak Accept)**

**Confidence:** 4

**Review:**

## Strengths

* **Clear problem framing**: focuses on *query design* for identifiability, not just training tricks.
* **Mechanistic intuition**: ties success to **variance of pre-activations** and avoiding activation saturation; provides histograms/variability plots.
* **Concrete metrics for reconstruction**: fraction of matched neurons (*m/r*) and cosine distances between teacher/student weights and output heads; tabulated across many settings.

## Weaknesses

* **Matching protocol unspecified**: How are student and teacher neurons paired before reporting cosine distances and *m/r*? (Hungarian assignment? Thresholding?) Without this, reconstruction metrics are ambiguous and possibly optimistic. Please specify and release code.
* **Selective generalization checks**: Fashion-MNIST losses are used as a proxy, but the goal is **weight recovery**, not cross-dataset accuracy. Add *within-distribution* generalization (MNIST-test) alongside **exact recovery** metrics to separate "functional imitation" from "identifiability". Some plots exist but are not summarized numerically.
* **Grid-Composition realism**: 3×3 patch stitching induces **out-of-manifold** images. That may be fine for probing, but can also **bias EC dynamics** toward degenerate directions. Provide an ablation comparing *random patch selection* vs *class-consistent composition*, and quantify how often grid samples cross multiple teacher hyperplanes (the claimed mechanism).
* **Theory lightness**: The link "more pre-activation variance $\Rightarrow$ more identifiability" is argued empirically. A short lemma for a 1D neuron (identifiability from $\ge 3$ points in the non-saturated regime) and a high-dimensional *measure* of hyperplane-crossing probability would greatly strengthen claims.
* **Scope of generality**: Everything targets a **known activation** $g(x)=\text{softplus}(x)+\sigma(4x)$ to break symmetries. Discuss robustness to **mismatch** (teacher ReLU / student g, or unknown slope), noise on outputs, and label quantization. Edge-case in Appendix shows pre-activations can mislead for ReLU; quantify how often.
* **Reproducibility**: No **code link**, incomplete details (seed control, exact selection of 60k grid comps from $D^9$ space, augmentation magnitudes schedule, EC hyperparams per setting). List exact seeds and release scripts to reproduce Tables/Figures.
* **Claim calibration**: "recover up to **100×** more parameters than samples" is compelling; please report the *exact* successful teacher parameter count and not just hidden size (accounting for input $\to$ hidden, biases, hidden $\to$ output) for the 5k setting, and add CIs across $\ge 3$ seeds.

## Correctness & Technical Soundness

* The **EC** setup and imitation loss are standard; training/test loss discrepancies convincingly signal overfitting with weak augmentations. However, the **success** cases rely on **very small cosine errors**, ensure these are *post-permutation and sign/scale symmetries fixed* and computed on **normalized vectors**. Clarify whether output-weight distances are evaluated after aligning hidden units.
* The **pre-activation variability metric** ("% std increase") is intuitive but under-defined. Provide the exact formula, aggregation across neurons, and whether centering is per-neuron or global.

## Evidence & Validation

* Table 2 shows strong numbers for **$\text{MNIST} \pm \eta[0,1]$** and **$\text{Grid-Comp.} \pm \eta[0,1]$**, including *m/r* = 1.00 for **512**, excellent if matching is correct. Complement with **confidence intervals** over seeds and **sensitivity** to noise magnitude (you partly vary [0,0.5], [0,1], [0,2], summarize the stability range explicitly).
* Provide at least **one ablation** isolating each component: (i) **biased-noise only**, (ii) **grid only**, (iii) **both**, with the same Q to attribute gains cleanly (some results are separated across figures/tables but not normalized to the same Q).

## Reproducibility Signals

* Appendix lists hardware, EC params, and optimizer/scheduler. Still needed: **code**, **random seeds**, **exact grid-sampling recipe**, **counts per augmentation type**, **criteria for declaring a neuron "reconstructed"**, and complete **metric definitions**.

## Questions for Authors

1. **Neuron matching**: Describe the matching algorithm and symmetry resolution used to compute *m/r* and cosine errors. Provide code.
2. **Exact "100×"**: Report teacher parameter counts for each successful setting with **5k** base samples, with seeds/CIs.
3. **Grid sampling policy**: How are the 60k grid composites chosen from $D^9$? Any class constraints or duplicate avoidance?
4. **Activation robustness**: What happens if the teacher is ReLU, GELU, or $g$ with a misspecified slope? Include at least one mismatch experiment.
5. **Active vs. passive**: Compare to an active query method that directly maximizes pre-activation changes; report cost vs. your augmentations.

**Score:**

3

**Topic Fit:**

3

---

### Official Review · Reviewer_xz7i · 2025-09-15
**Paper needs rewriting and clearly highlighting novelty**

**Confidence:** 3

**Review:**

The paper attempts to reverse engineer network weights of a model on MNIST dataset. They empirically show that the standard augmentation techniques do not improve reconstruction. However, their proposed methods are able to retrieve the weights with higher accuracy for networks up to 100 times the training points.

## Strengths
The paper explores an interesting problem.

## Weakensses
- The writing is hard to follow. I think it is good to show what did not work. But perhaps a discussion section is better suited for this. It may help the reader if the authors focus on their main contributions in a separate methods section and present the effect of traditional augmentation techniques in the experiments/results section as baselines. I think some rewriting will significantly improve the paper.
- It seems unusual for the reconstruction error to fluctuate so much from 2^8 to 2^9. I wonder if this is due to stochasticity. Perhaps the authors can present error bars over multiple runs rather than line plots.
- The paper refers to Martinelli et al., 2024, but it will help if the authors explain clearly how their approach fits within the framework of Expand-and-Cluster.
- Results on a few more datasets will strengthen the paper.

**Score:**

2

**Topic Fit:**

3

---

### Official Review · Reviewer_tNqu · 2025-09-16
**Pre-activation variability offers valuable insights, but the techniques may require adaptation for modern applications**

**Confidence:** 4

**Review:**

The extended abstract addresses neural network reverse engineering in teacher-student setups, where the goal is to reconstruct a teacher network's weights by training student networks on input-output queries. The authors identify a critical limitation: when teacher networks have more parameters than available training data points, standard approaches fail because students overfit to the specific queries rather than learning the underlying function. They develop two novel data augmentation techniques - biased noise and grid composition - specifically designed to increase variability in teacher pre-activations. Their methods claim to successfully reverse-engineer networks 100x larger than the training dataset, significantly extending previous capabilities that were limited to 256-neuron teachers.

Strengths:
1. The paper pinpoints pre-activation variability as the core issue in neural network reverse-engineering.
2. The authors proposed a principled approach to augmentation design. The biased noise technique (±η[0,1]) creates systematic directional perturbations that better explore weight vector directions, while grid composition generates structured synthetic data that maintains distributional properties.
3. They achieve reverse-engineering for networks 100x larger than the training data represents a substantial improvement over prior work.

Weaknesses:
1. The work focuses exclusively on single-layer feedforward networks with specific activation functions on MNIST data. The scalability to deeper complex networks, different architectures, or other domains remains unclear.
2. The choice of asymmetric activation function g(x) = softplus(x) + sigmoid(4x) may not reflect realistic scenarios with standard activation functions and unknown architectures.
3. Doesn't discuss the computational costs of their augmentation techniques. Grid composition can generate D⁹ datapoints, which could be computationally prohibitive for larger base datasets.
4. Testing only on MNIST, while common for proof-of-concept work, limits confidence in the generalizability. The techniques may not work as effectively on more complex datasets with different statistical properties or higher-dimensional natural images.

**Score:**

4

**Topic Fit:**

3

---

### Official Review · Reviewer_oHro · 2025-09-16
**Interesting work with room for improvement.**

**Confidence:** 4

**Review:**

Summary:
The paper proposes usage of a data augmentation method for model stealing, i.e. approximating the weights of teacher neural network model. Using and building upon the expand-and-cluster framework, the authors expand its capability to learn the weights of teacher model up to 512 neurons per layer, which is double the size of the baseline. They compare the impact of their augmentation method with traditional augmentations such as flipping and rotation, claiming that it is more effective.


Strengths:
* The augmentation is simple and straightforward to carry out.
* The scale of 100x for the teacher model could be significant.

Weaknesses:
* My main question is about the strategy used for the augmentation. Given that 9 cuts from other images are combined to create a new image, I’m wondering if it introduces noisy and non-existent inputs for the student model?
* In addition, for comparison some simple augmentation methods have been utilized. I wonder how the method compares to more complex ones such as image occlusion/obfuscation or generative augmentations?
* Finally, Figure 2 shows that grid-comp plus noise augmentation is at its minimum loss with 512 layer size, so it makes sense to continue this experiment to find out about where it breaks.

**Score:**

3

**Topic Fit:**

2